# Stimulus presentation can enhance spiking irregularity across subcortical and cortical regions

**Saleh Fayaz**[1☯], **Mohammad Amin Fakharian**[1,2☯], **Ali Ghazizadeh**[1]*

**1** Electrical Engineering Department, Sharif University of Technology, Tehran, Iran, **2** School of Cognitive Sciences, Institute for Research in Fundamental Sciences, Tehran, Iran

☯ These authors contributed equally to this work.
* alieghazizadeh@gmail.com

**Data Availability Statement:** The data related to vlPFC and SNr and the code used in this work is deposited in Dryad doi:https://doi.org/10.5061/dryad.0cfxpnw2c. Analyzed data for V1, V2, MT, PMd and IpN are from publicly available sources as

## Abstract

Stimulus presentation is believed to quench neural response variability as measured by fano-factor (FF). However, the relative contributions of within-trial spike irregularity and trial-to-trial rate variability to FF fluctuations have remained elusive. Here, we introduce a principled approach for accurate estimation of spiking irregularity and rate variability in time for doubly stochastic point processes. Consistent with previous evidence, analysis showed stimulus-induced reduction in rate variability across multiple cortical and subcortical areas. However, unlike what was previously thought, spiking irregularity, was not constant in time but could be enhanced due to factors such as bursting abating the quench in the post-stimulus FF. Simulations confirmed plausibility of a time varying spiking irregularity arising from within and between pool correlations of excitatory and inhibitory neural inputs. By accurate parsing of neural variability, our approach reveals previously unnoticed changes in neural response variability and constrains candidate mechanisms that give rise to observed rate variability and spiking irregularity within brain regions.

## Author summary

Mounting evidence suggest neural response variability to be important for understanding and constraining the underlying neural mechanisms in a given brain area. Here, by analyzing responses across multiple brain areas and by using a principled method for parsing variability components into rate variability and spiking irregularity, we show that unlike what was previously thought, event-related quench of variability is not a brain-wide phenomenon and that point process variability and nonrenewal bursting can enhance post-stimulus spiking irregularity across certain cortical and subcortical regions. We propose possible presynaptic mechanisms that may underlie the observed heterogeneities in spiking variability across the brain.

noted below: V1-2 data: https://crcns.org/data-sets/vc/v1v2-1/about_v1v2-1 IpN data: https://crcns.org/data-sets/cb/cb-1/about-cb-1 MT and PMd data: https://churchland.zuckermaninstitute.columbia.edu/content/code.

**Funding:** The author(s) received no specific funding for this work.

**Competing interests:** The authors have declared that no competing interests exist.

## Introduction

While mean firing rate is widely used as a proxy of neural communication code, firing rate variance is also shown to play a significant role in neural coding and to serve as a diagnostic tool for distinguishing underlying neural mechanisms [1–7]. In particular, neural response variability is believed to change during development and to be correlated with behavioral performance, stimuli conditions and to vary across different neural states [8–12]. Several studies have shown that spike count variability changes across different conditions in several cortical, subcortical, and cerebellar regions [1,3,4,10,13–17]. Importantly, numerous observations support a drop in neural response variability as measured by fano-factor (FF) in response to sensory or motor stimuli across cortical regions [3,6,18]. The variation in neural spike count when viewed as a doubly stochastic process may be due to two components parsed by the law of total variance (1) the spike count variability within the trials, (2) the variability of spiking parameters between the trials. The first component will be referred to as within trial spiking irregularity (SI or $\Psi$) and gives rise to the expected variance of the count (EVC) while the second component can be driven by between trial rate variability (RV) and forms the variance of the expected count (VEC) [1,4,19]. Despite this general understanding, a principled approach for estimating these two components and tracking their concurrent changes in time is not available. Consequently, the relative contributions of $\Psi$ and RV to the widely reported post-stimulus reduction in FF in the brain is not known.

A previous attempt to estimate the RV component [4] (VarCE method) showed that FF reduction is paralleled by the reduction in VEC (VarCE is an estimate of VEC). However, this approach assumed $\Psi$ to be related to firing rate by a constant coefficient $\phi$ in time (see S1 Text). Other more recent methods also followed the same assumption for spike count variability decomposition [1,19]. Clearly, such an assumption can result in gross misestimations of EVC and VEC (or equivalently $\Psi$ and RV) if $\phi$ also changes in time. Even in the absence of such changes, VarCE only estimates the relative changes in VEC across time and not its absolute value. Thus, the true nature and accurate estimation of $\phi$ remains unaddressed.

Here, we present a principled approach that allowed us to find the absolute value of VEC and track changes in $\phi$. Briefly our approach works by explicitly relating FF to normalized RV and $\Psi$ (nRV and n$\Psi$). n$\Psi$ which is the same as $\phi$ is estimated by invoking minimal assumptions for a doubly stochastic point process and by adapting an existing method that allowed relatively accurate estimation of n$\Psi$ [19]. Furthermore, it is shown that while n$\Psi$ is robust to rate non-stationarities in time, it can be inflated beyond point process variability given certain violations of renewal model such as bursting.

Indeed, results show that unlike what was previously assumed, n$\Psi$ could show robust changes during the stimulus presentation across several subcortical and cortical regions. However, while the post-stimulus nRV showed a ubiquitous reduction across all areas examined, post-stimulus n$\Psi$ change was more heterogenous: decreasing in some (mostly cortical) areas and increasing in others (mostly subcortical). Moreover, while we found the post-stimulus reduction in FF to be mostly due to nRV decrease, the concurrent change in n$\Psi$ was shown to convey critical information regarding the changes in the spiking pattern within a trial such as occurrence of bursting. Our results caution against strong claims about post stimulus response variability quench across the brain. The heterogeneity in patterns of nRV and n$\Psi$ during an event constrains and contrasts the underlying neural mechanisms across different regions.

## Results

### Parsing neural response variability components

A stochastic process with deterministic parameters would induce variability in the spike count, i.e. point-process variability. Moreover, stochastic selection of such parameters between different trials would impose variability across trials (doubly stochastic process). Let's assume a doubly stochastic process in which inter-spike interval (ISI) parameters change probabilistically across trials. In this case, spike count variability can be decomposed into at least two components using the law of total variance in the following form:

$$\mathrm{Var}(N_T) = \underbrace{\mathrm{E}\left[\mathrm{Var}\left(N_{T_i}|\gamma_i\right)\right]}_{\text{EVC}} + \underbrace{\mathrm{Var}\left(\mathrm{E}\left[N_{T_i}|\gamma_i\right]\right)}_{\text{VEC}} \tag{1}$$

Where $N_T$ denotes the spike count in the time-bin ($T$) and $\gamma_i$ represent ISI parameters in trial. Here the first term denotes the expected variance of spike count (EVC) due to spiking irregularity ($\Psi$) within a trial given constant parameters, and the second term denotes variance of expected count (VEC) due to rate variability (RV) across trials. Total count variance or Var ($N_T$) is related to FF via a simple normalization by $\mathrm{E}[N_T]$. Eq 2. Reveals the contribution of $\Psi$ normalized by $\mathrm{E}[N_T]$ (n$\Psi$) and RV normalized by $T \times \mathrm{E}[N_T]$ (nRV) to the total FF as following:

$$\mathrm{FF}(N_T) = \frac{\mathrm{Var}(N_T)}{\mathrm{E}[N_T]} = \mathrm{n}\Psi + T \times \mathrm{nRV} \tag{2}$$

$$\mathrm{n}\Psi = \frac{\mathrm{E}[\mathrm{Var}(N_T|\gamma_i)]}{\mathrm{E}[N_T]}, \mathrm{nRV} = \frac{\mathrm{Var}(\mathrm{E}[N_T|\gamma_i])}{T \times \mathrm{E}[N_T]}$$

The motivation for defining n$\Psi$ and nRV in this way, comes by considering renewal processes which has been widely used as models of spiking neurons [20–29] for sufficiently large time-bins where we have:

$$\mathrm{n}\Psi = \lim_{T\to\infty} \frac{\mathrm{E}\left[\mathrm{Var}\left(N_{T_i}|\gamma_i\right)\right]}{\mathrm{E}[N_T]} = \frac{\mathrm{E}\left[\lambda_i^3 \sigma_i^2\right]}{\mathrm{E}[\lambda_i]}, \mathrm{nRV} = \lim_{T\to\infty} \frac{\mathrm{Var}\left(\mathrm{E}\left[N_{T_i}|\gamma_i\right]\right)}{T \times \mathrm{E}[N_T]} = \frac{\mathrm{Var}(\lambda_i)}{\mathrm{E}[\lambda_i]} \tag{3}$$

Where $\lambda_i = \frac{1}{\mathrm{E}[\mathrm{ISI}_i]}$ and $\sigma_i^2 = \mathrm{Var}(\mathrm{ISI}_i)$ and $\mathrm{ISI}_i$ is the random variable representing the inter-spike intervals drawn from a given distribution whose parameters can vary across trials (see S1 Text for details). Simply put while n$\Psi$ affects the patterning of spikes resulting in more periodic or more variable spike times, nRV is concerned with variability in the overall intensity of spiking across trials (S1 Fig).

According to Eq 3, FF is asymptotically a linear function of time-bin. It includes an intercept which represents normalized within trial spiking irregularity (n$\Psi$ aka $\phi$) and a slope which represents normalized between trials rate variability (nRV). For a simple stochastic point process with no trial-to-trial variation in ISI parameters, FF should be constant for sufficiently large time-bins. Fig 1a shows that indeed after a short transient, FF converges to the theoretical prediction based on the parameters of the ISI distribution (i.e. it is 1 for Poisson point process and is $\frac{1}{\kappa}$ for Gamma($\kappa, \theta$)). On the other hand, for a doubly stochastic process with trial-to-trial variation in rate, FF increases in an asymptotically linear fashion as a function of time-bin. Note that in this case the intercept of the line which represents n$\Psi$ remains unchanged as long as the spiking irregularity ($\kappa$ in Gamma($\kappa, \theta$)) remains unchanged across trials (Fig 1a). It can be shown that for a general renewal process n$\Psi$ is equal to the normalized

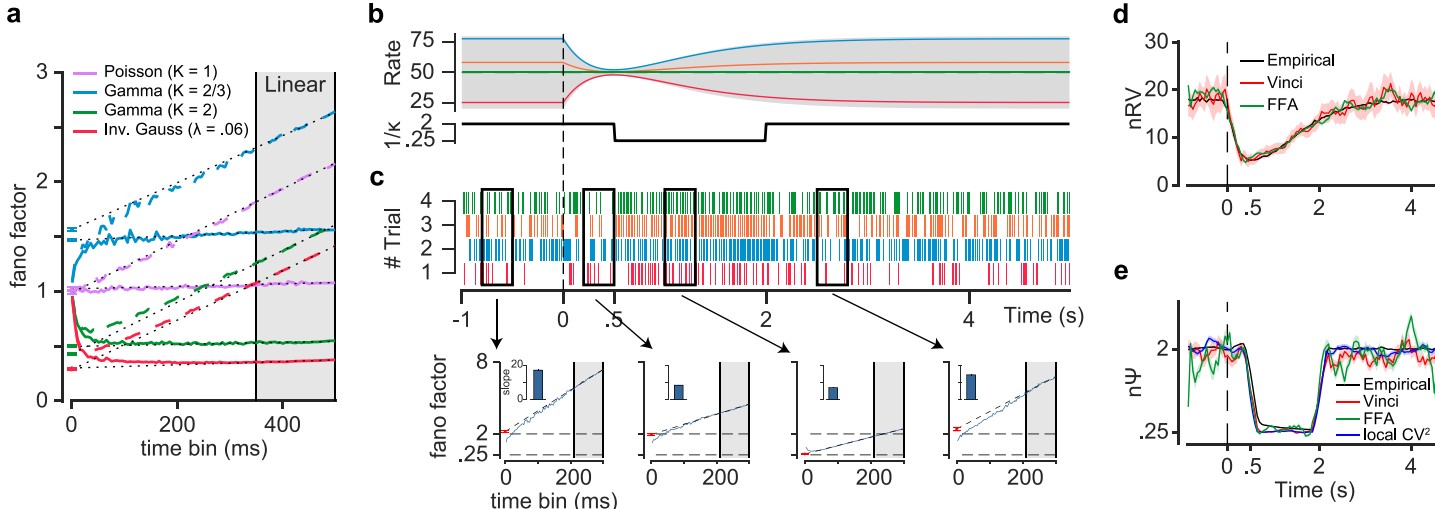

**Fig 1. Parsing components of variability in doubly stochastic renewal processes in time.** (a) Fano factor (FF) as function of time-bin for simple (solid lines) and doubly stochastic (dashed lines) renewal processes with different ISI distributions (exponential, gamma and inverse gaussian). The asymptotic part of FF for large time-bins (gray shaded area) is used for independent estimates of normalized spiking irregularity (nΨ) as the intercept and the normalized rate variability (nRV) as the slope. Note the non-zero slopes for doubly stochastic processes ($\lambda_i \sim N(\mu = 50Hz, \sigma = 10Hz)$). (b) Rate parameter ($\lambda = \frac{1}{\kappa\theta}$) and inverse of shape parameter ($\frac{1}{\kappa}$) for a doubly stochastic gamma process as a function of time. Rate parameter was variable across trials according to $50 + n_i \times \left(1 - 5t_+e^{-2t_+}\right)$ where $n_i \sim N(0, 30)$ and $t_+$ is time for positive values and zero otherwise. (c) top, raster plot shows multiple trial realizations from the stochastic gamma process (rate of each trial is indicated in panel b), bottom, FF as a function of time-bin (0-300ms), blue curve represents FF estimates as a function of time. Dashed line shows extrapolation of fitted line to the linear portion of FF; red dot represents intercept. Slopes of each line are indicated by bar plots. (d) nRV estimates made empirically by assuming access to a large number of realizations of each trial with given parameters (black, empirical estimates see S1 Text) along with estimates using smoothed FFA (green) and Vinci (red) methods. (e) same format as d but for nΨ estimates. Blue trace represents $CV^2_{local}$. Error bars and shades indicate the sem.

point process variability (nPPV) which is measured by $CV^2_{local}$ (which can be derived from $CV2$, see S1 Text) and shown to provide robust and unbiased estimation of $\kappa^{-1}$ [13,23,30–33] (not to be confused with coefficient of variation or $CV^2$ which is essentially the same as FF in the limit of large T). Notably, this equivalence holds for any renewal process regardless of the shape of ISI distribution as illustrated for exponential, gamma and inverse gaussian probability density functions (Fig 1a, for general proof see S1 Appendix). Also note that while in theory FF asymptotic behavior is at infinity limits of time-bin, in practice the linear asymptotic behavior may be observable with much smaller time-bins for neurons with sufficiently high firing rates (S2 Fig).

Indeed, by examining the FF asymptote (FFA), one would be able to track the temporal fluctuations in nΨ and nRV over time using a sliding time-bin. Fig 1b shows a simulated doubly stochastic spike process with concurrent variations in Ψ and RV across time. Each trial was generated from a renewal point process with ISIs following a two-parameter Gamma($\kappa$, $\theta$) distribution. The process had two sources of variability: Firstly, the rate in each trial was 50Hz across trials with a variance that changed during the trial (Fig 1b top). This variation of rate around the 50Hz mean is responsible for RV. Secondly, in each trial, spikes were generated with the variability inherent to a gamma point process. In this case the parameter $\kappa$ is inversely proportional to spiking irregularity and was set to change in time (e.g. temporal changes in $\frac{1}{\kappa}$ from 2 to 0.25 and back to 2). Fig 1c shows spiking realizations of this doubly stochastic gamma process across four example trials each corresponding to a given rate dynamic. Interestingly, FF intercept and slope in a given 300ms time-bin closely paralleled the expected changes in nΨ and nRV during the trial. That is, the slope showed a reduction concurrent with the reduction of rate variability after time zero while the intercept independently followed

changes in $\frac{1}{\kappa}$ that was delayed (onset asynchrony) compared to RV (going from 2 to 0.25 and back to 2).

In practice, fitting the asymptotic behavior of FF, while intuitive, may not lead to minimum variance estimators of nΨ and nRV (e.g. nΨ estimates can be noisy as small changes in slope translate to large swings in the intercept). Therefore, we used a more robust estimator of EVC using an existing nonparametric approach developed by Vinci et al 2016 (referred to hereafter as the Vinci method, see S1 Text) [19]. Once the EVC is estimated one can normalize it by $\langle N_T \rangle$ which is the average of $N_T$ across trials (as one's best estimate of $E[N_T]$) to estimate nΨ. Given FF and nΨ one can then estimate nRV using Eq 2. This method allows for a more robust estimation of nΨ and nRV (Fig 1d and 1e). Note that both FFA and Vinci methods require sufficiently large number of trials and spikes per time-bin to give accurate estimates of nΨ and nRV. Obviously, the choice of the optimal time-bin also depends on how fast variability components change in time (faster changes favor smaller time-bins to maintain sufficient temporal resolution, S3 Fig) similar to any other time-histogram method [34,35]. However as noted, the Vinci estimate is more robust compared to FFA across a range of parameters (S2 Fig) and will be mainly used when analyzing neural data in this paper.

While Vinci and FFA estimates are obtained by having access to a single realization for each rate pattern similar to the conditions of dealing with real neural data, in the case of simulations one can afford to generate a large number of spiking rasters per each combination of rate and $\kappa$ to estimate nΨ and nRV (referred to as empirical estimate, see S1 Text). Such empirical estimates give the best possible estimate for nΨ and nRV any method could hope to acheive and serve as a benchmark to gauge a method's performance on simulated data. As can be seen, the empirical estimates show a good agreement with estimates using Vinci and FFA methods (Fig 1d and 1e). Furthermore, $CV^2_{local}$ which measures the inherent randomness of spike generation or nPPV also matches the nΨ estimate both tracking the changes in $\frac{1}{\kappa}$ (Fig 1e).

## Decomposition of nΨ

Simulation results show that for renewal point processes, nΨ will be the same as nPPV measured by $CV^2_{local}$ regardless of particular ISI distributions used (Fig 2a and 2b), between trial rate variability captured by nRV (Fig 2c) or firing rate non-stationarities during a trial (Fig 2d and 2e). Both FFA and Vinci methods track the ideal values of nΨ and nRV with good accuracies in such cases (Fig 2a and 2e). On the other hand, bursting [30] which is a clear violation of the renewal assumption (i.e. history dependance to previous spike occurrences beyond ISI parameters) can be shown theoretically to inflate nΨ beyond nPPV (S1 Appendix) and accordingly results in larger spiking irregularity beyond nPPV (Fig 2f). Notably, such inflation of empirical nΨ is not observed even when there is significant non-stationarities such as rate noise [36] or rate switching [7,37] (Fig 2d and 2e) consistent with theoretical predictions (S1 Appendix) [38]. Note however that while Vinci and FFA tolerate random rate noise, they seem to slightly overestimate nΨ in rate switching with random and rapid changes from low to high firing across trials.

The fact that nΨ is not simply distorted by firing rate fluctuations during a time-bin is fortunate for our methods which require relatively large time-bins during which firing rates might change. Additional simulations confirm that as long as the nPPV remains unchanged our estimates of nΨ is largely robust to non-stationarities during a time-bin (S4 Fig). Thus, one may decompose nΨ into $CV^2_{local}$ which measures point process variability (nPPV) and $n\Psi - CV^2_{local}$ as a measure of spiking irregularity by sources such as bursting which is shown to have multiplicative and additive effects to increase nΨ (S1 Appendix).

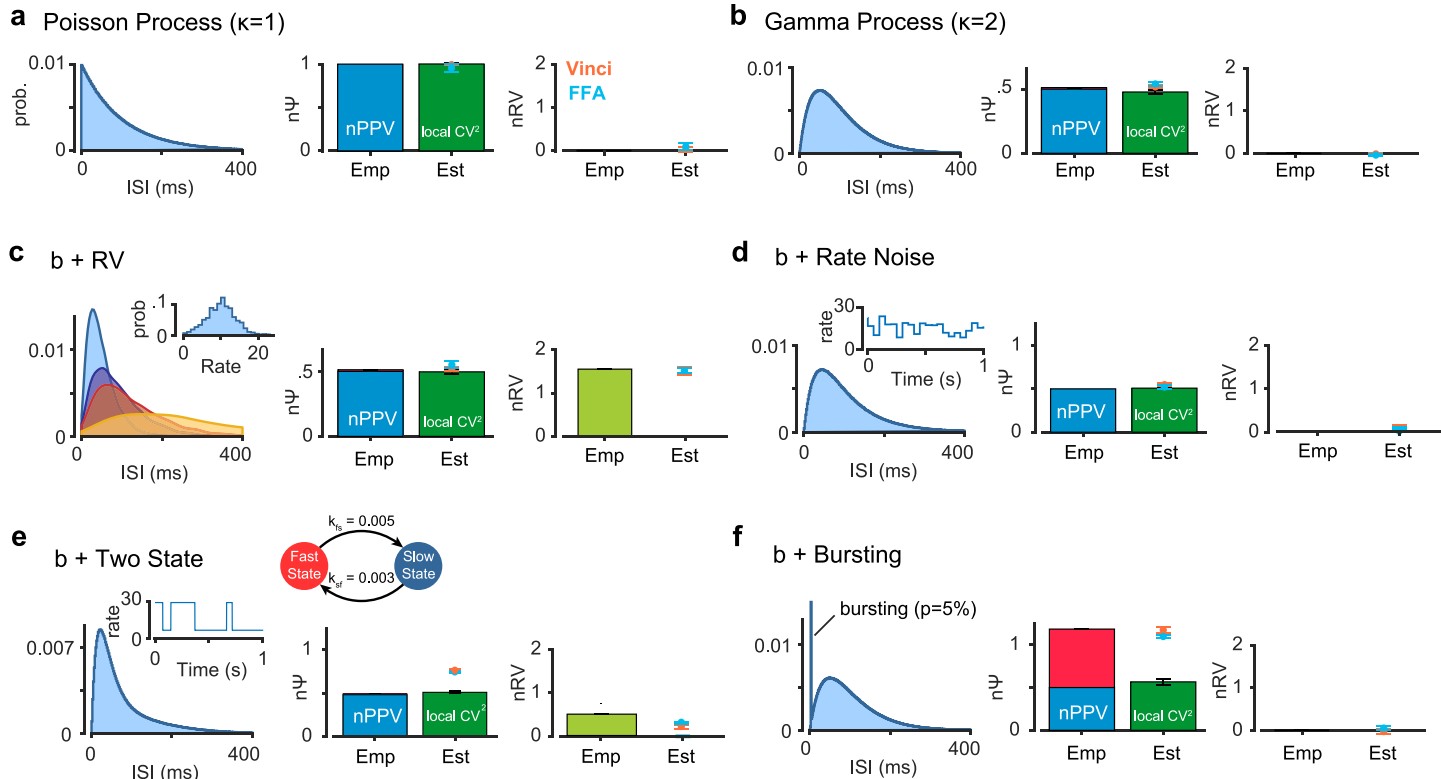

**Fig 2. Normalized point process variability (nPPV) and bursting contribute to spiking irregularity (nΨ).** Empirical estimations (by generating several samples from each trial, see S1 Text) of nΨ and nRV along with estimations made by FFA and Vinci methods. Theoretical nPPV and its estimate using $CV^2_{local}$ is also shown. The red box represents total empirical nΨ; blue box represents theoretical nPPV ($1/\kappa$); dark green box represents estimated $CV^2_{local}$; light green box represents empirical estimate of nRV; orange and blue dots represent Vinci and FFA estimates for nΨ and nRV in the corresponding plots; errorbars are sem. (a) Poisson process with rate 10Hz with no rate variability. (b) Gamma process ($\kappa = 2$) with no rate variability. (c) same as b but doubly stochastic with rate variability. (d) same as b but with within trial white noise added to the constant rate ($\sim N(0,3)$ and $\Delta t$ 50$ms$, see S1 Text). (e) same as b but with within trial state switching between high and low-rates imposed (see S1 Text). (f) An example ISI distribution with bursting imposed with probability 0.05 on a given spike; during generation of the process, each spike was replaced by 5 consecutive spikes with 3ms ISI.

## Insufficiency of estimates that have a constant φ assumption

Given changes in nΨ, methods that assume a constant $\phi$ (i.e. constant nΨ or spiking irregularity in time) can misestimate nRV. For instance, the VarCE method which attempts to estimate VEC by finding the largest constant $\phi$ (nΨ) that keeps VEC positive during the trial, can become error prone:

The VarCE method is also based on parsing sources of variability using the law of total variance in which VarCE and $\phi$ here are measures of VEC and nΨ, respectively.

$$\text{VEC} = \text{VarCE} = \text{Var}(N_T) - \phi\text{E}[N_T] \tag{4}$$

Note that even in cases when $\phi$ is constant, VarCE can only measure VEC up to additive shifts (S5a Fig). More importantly, if the true $\phi$ changes and in the absence of changes in VEC, VarCE erroneously reports changes in VEC (S5b Fig).

## Post-stimulus enhancement of nΨ and reduction of nRV in cerebellum

Cerebellar neurons show sizeable changes in their variability in response to different conditions [13,39–41]. We used our method to parse rate variability and spiking irregularity across

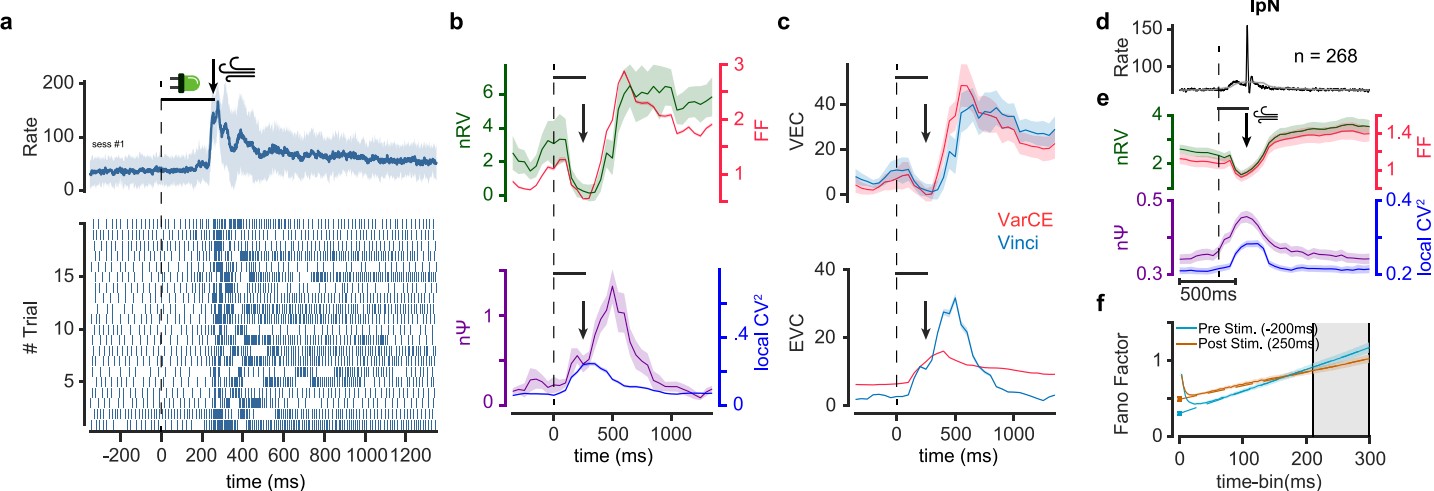

**Fig 3. Post-stimulus changes in rate variability and enhancement of spiking irregularity in cerebellum.** (a) Example IpN neuron PSTH with standard deviation as the shaded area (top) and raster plot of 20 example trials (bottom) aligned to the conditioned stimulus (CS) onset (LED light) followed by unconditioned stimulus (US, airpuff) in 250ms. (b) Post-stimulus FF and nRV showing an initial decrease followed by sustained increase (top) and post-stimulus nΨ showing a robust increase (bottom). $CV_{local}^2$ also shows a moderate increase following stimulus onset. (c) VEC and EVC estimations using the Vinci vs VarCE method (top and bottom, respectively). VarCE overestimates the rise in VEC (related to rate variability) and underestimates EVC (related to spiking irregularity). (d) Average PSTH of IpN neural population (n = 268) neurons. Black and gray traces show PSTHs with 10ms and 300ms bin-widths, respectively. 300ms bin-width is equal to time-bin used for variability estimation in the Vinci and FFA methods. (e) Average FF and nRV (top) and nΨ and $CV_{local}^2$ (bottom) as a function of time within the sliding time-bin (300ms, 50ms sliding). Dashed line indicates CS onset, horizontal line indicates CS on period and arrow indicates US onset. (f) FF as a function of time-bin centered on pre- and post-stimulus onset periods (-200 and 250ms respectively). Gray area indicates the linear zone at which asymptotic FF is estimated. The increase in intercept and the decrease in slope is consistent with changes in nΨ and nRV respectively. Error bars and shades are sem.

neurons (n = 268) recorded from Interposed Nucleus (IpN) of mice during the Pavlovian eyeblink conditioning task [40]. Fig 3a shows peristimulus time histogram (PSTH) response of an example IpN neuron time locked to the conditioned stimulus (CS) onset (LED light). 250ms after the CS onset the unconditioned stimulus (US: air-puff) was delivered. As can be seen the example neuron showed strong excitation following CS onset and just prior to the US delivery. Fig 3b top shows an initial decrease in the average FF followed by a sustained increase. As can be seen, this temporal pattern was largely due to very similar pattern of change in rate variability (nRV). This pattern of change in nRV and in particular its rise after about 500ms from CS can be easily verified by looking at the spiking raster plot (Fig 3a bottom). Interestingly, despite the initial decrease in nRV and FF, nΨ showed a robust post stimulus increase. The increase in nΨ was only partially due to the increase in nPPV as measured by $CV_{local}^2$ pointing to extra factors elevating the observed spiking irregularity which was predicted previously.

In the cerebellar neural example, VarCE estimated the trends of change in VEC accurately but somewhat overestimated it particularly during the time when nΨ was changing (Fig 3c top). More importantly, since estimated $\phi$ in VarCE was constant, it grossly misestimated the component of variability that was due to spiking irregularity VEC (i.e $\phi E[N_T]$) (Fig 3c bottom).

Similar to the neural example shown, the pattern of FF reduction and subsequent increase was also observable across the cerebellar IpN population and seemed to be driven primarily by the temporal dynamics in nRV. Once again during the same period nΨ showed a robust post stimulus increase which was accompanied by concurrent increase in nPPV which was nevertheless only partially explaining the increase observed in nΨ. The post stimulus reduction in nRV and increase in nΨ can also be verified using the FFA method by looking at the linear

asymptotic behavior of FF which showed intercept increase and slope decreased after the stimulus onset.

## Post-stimulus enhancement of nΨ canceling reduction of nRV in substantia nigra

Next, we checked response variability in another subcortical area namely substantia nigra reticula (SNr). This region was particularly informative for our analysis since here individual neurons showed large dynamic range of excitatory or inhibitory responses to visual objects based on their past value memory (excitation to low value objects and inhibition to high value objects, [42]). Fig 4a and 4b show responses of example neurons to high value and low value objects, respectively. For the neuron with inhibitory response a post stimulus reduction in FF was observed as expected. Similar to what was observed in cerebellum this decrease was largely due to a drop in post-stimulus rate variability (nRV) rather than spiking irregularity (nΨ) which instead showed a robust post stimulus increase (Fig 4c). Interestingly, the neuron with excitatory response also showed a drop in nRV. However, this drop was more than offset by the increase in the spiking irregularity such that post stimulus FF in this neuron did not show the expected quench in spiking variability (Fig 4d). Once again, the radical changes in nΨ in both neurons meant that the estimates of VEC using VarCE were erroneous (Fig 4e and 4f). Moreover, there was a gross misestimation of EVC using VarCE especially in the case of the neuron with the inhibitory response. This is because constant $\phi$ assumption (or equivalently constant nΨ), inevitably predicts a drop in Ψ for inhibitory responses which is the opposite of what was observed for this neuron (Fig 4e).

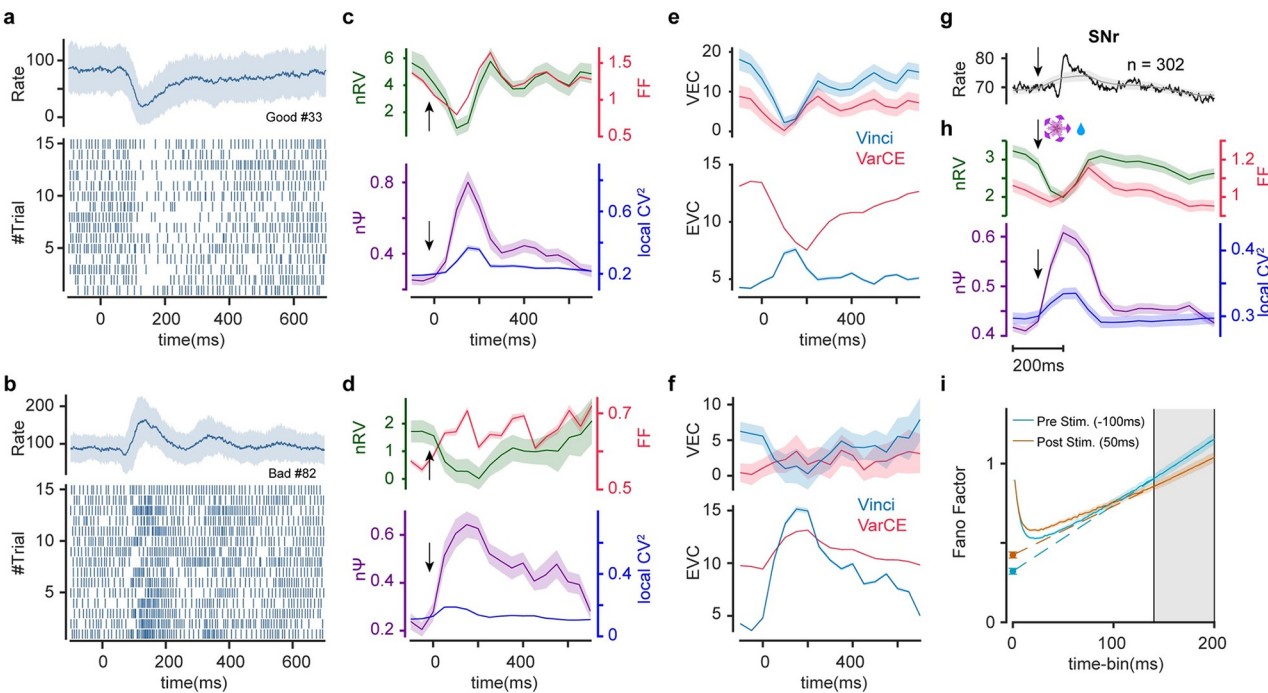

**Fig 4. Post-stimulus drop in rate variability is compensated by concurrent rise of spiking irregularity in SNr.** (a-f) Same format as Fig 3a and 3c but for one example SNr neuron showing inhibition (a, c, e) and another example SNr neuron showing excitation (b, d, f) to object onset. Raster plots show 15 sample trials for each neuron. (g-i) Same format as Fig 3d and 3f showing population average of 302 SNr neurons. Black and gray traces show PSTHs with 10ms and 200ms bin-widths, respectively. Sliding time-bin of 200ms with 50ms steps were used for estimators of spiking variability.

Across the population of recorded neurons, stimulus onset was associated with a drop in nRV, on the other hand, nΨ showed a robust increase in response to the stimulus onset revealing enhanced spiking irregularity (Fig 4g and 4h). Similar to the cerebellar neuron, population $CV^2_{local}$ also increased following the stimulus onset suggesting a concurrent increase in nPPV which nevertheless again failed short of explaining the larger increase in nΨ (Fig 4h). The decrease in nRV and increase in nΨ can be visualized by the FFA method as well (Fig 4i). Interestingly, unlike the cerebellar IpN neurons, due to sizeable increase in nΨ, dynamics of the FF and nRV was dissimilar for this region (S6 Fig). Indeed, SNr is one brain region which also violates presumed reduction of post stimulus FF [3].

As shown in Eq 2, these results were obtained using normalized measures of between and within trial variabilities to control for firing rate change during the stimulus. Nevertheless, and to ensure robustness of our findings to firing rate changes during the trial as a possible confounder, we validated our results using two other approaches: (1) by using the mean-matching method which chooses subsets of neurons at each time point with the constraint of similar rate distributions [3] and (2) by repeating the analysis separately for neurons in different response types of excitatory, inhibitory and null post-stimulus responses. Both scenarios resulted in essentially the same behavior for nΨ and nRV and relatively stable FF during the trial for the SNr population (S7 Fig).

## Stimulus evoked changes in nΨ and nRV across cortical areas

It was previously shown that stimulus onset tends to reduce variability in cortical neural responses [3]. However, as stated previously due to lack of reliable methods for parsing the sources of spiking variability, it is not known how nΨ and nRV are affected by the stimulus onset. Here, we applied our methods on multiple cortical regions including areas V1, V2, middle-temporal (MT), premotor cortex (PMd) and ventrolateral prefrontal cortex (vlPFC) recorded in macaque monkeys by different groups [3,43,44]. As can be seen stimulus presentation or motor action caused a sizeable increase in average firing rate and a decrease in FF of all examined areas (Fig 5a); notice that for V1 and V2 the stimulation phase happens at the beginning of the trial and is turned off afterwards. In all cases the reduction in FF was concurrent with stimulus related reduction of nRV (Fig 5b). However, almost all areas showed concurrent changes in nΨ as well. With the exception of vlPFC, the spiking irregularity as measured by nΨ decreased during stimulus presentation across cortical regions. In vlPFC a transient post-stimulus increase in nΨ was observed. Note that despite the transient increase in nΨ in vlPFC, its FF was still quenched during stimulus presentation due to the larger size of nRV compared to nΨ (note the y-scales, S6 Fig). The changes in nΨ and nRV during stimulus on and off is also evident by looking at the intercept and slope of the FF in the time-bin (Fig 5c). Note that in regions with lower overall firing rate, larger time-bin is required for the asymptotic behavior of FF to emerge (compare 1000ms time-bin in V1 with 200ms time-bin in vlPFC). It is thus possible that for regions with low firing rate, fast temporal dynamics in nΨ and nRV are not detectable with low number of trials and large time-bins (such as transient increases similar to vlPFC). The relationship between nPPV and nΨ was more heterogenous between regions. While nPPV hardly changed in V1 and V2, it decreased during stimulus presentation in MT and vlPFC. Once again nPPV changes were relatively minimal and most of the changes in spiking irregularity was not due to nPPV (Fig 5b).

These results were once again corroborated using mean matching across the full population as well as separately within neurons showing excitatory, inhibitory or null responses across all the cortical regions (S8 Fig mean-matching and S9 Fig response clustering). In particular, the transient increase in nΨ in vlPFC was observed in all cases.

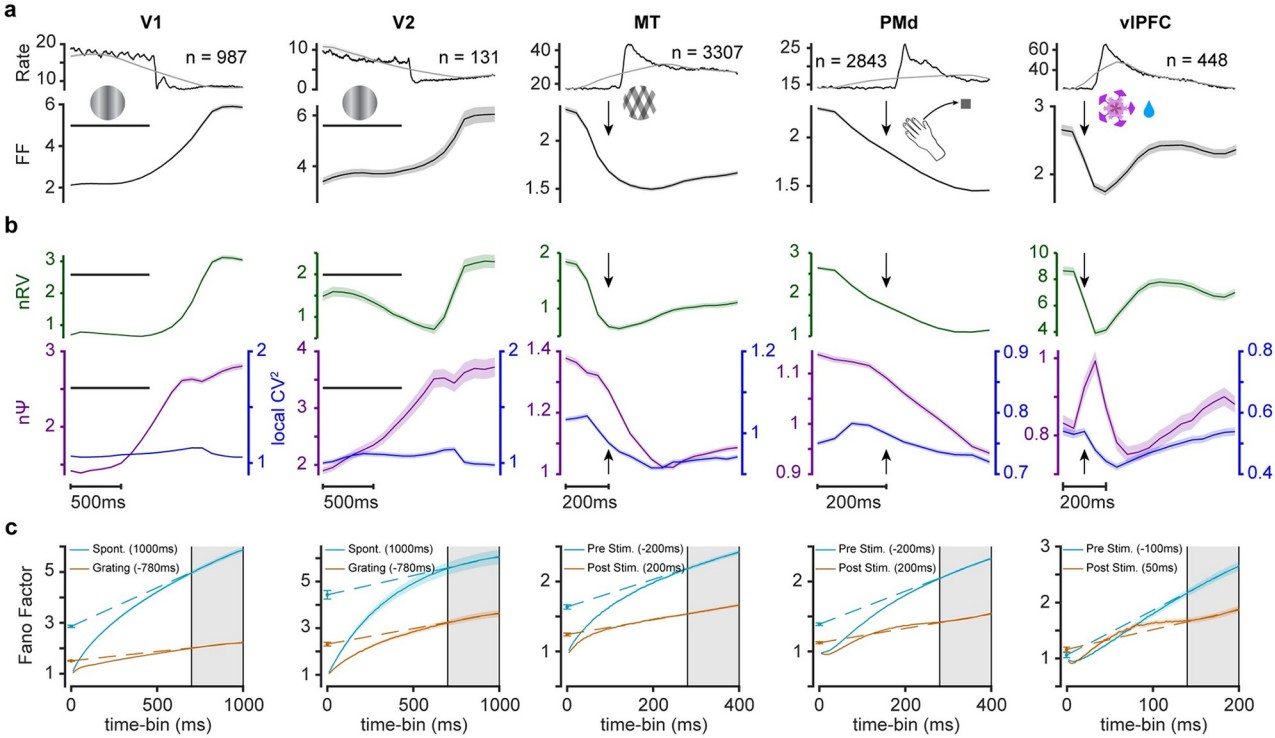

**Fig 5. Concurrent changes in rate variability and spiking irregularity during stimulus presentation or movement execution across the cerebral cortex.** (a) Average population PSTH across 5 different cortical regions responding to the stimulus presentation (areas V1, V2, MT, vlPFC) or movement execution (PMd). Black and gray traces show PSTHs with bin-widths equal to 10ms and matching time-bins used for estimation of FF and variability components in each region, respectively (time-bin = 1000, 1000, 400, 400, 200ms sliding by 100ms for V1/2 and 50ms for the rest respectively) (top) and FF dynamics for each region (bottom). Horizontal line indicates stimulus on period and arrow indicates stimulus onsets. (b) Concurrent temporal changes in nRV (top) and in nΨ (bottom) across cortical regions. Temporal dynamics of $CV^2_{local}$ is plotted alongside nΨ. (c) Same format as Fig 3f for each cortical area pre- and post- sensory or movement event.

To check whether, bursting is responsible for the observed inflation of nΨ beyond nPPV, the burst count per spike was detected and plotted alongside nΨ for each region (Fig 6). Interestingly, one observes similar temporal dynamics between nΨ and bursting in most regions (except for MT and PMd) such that often increases in bursting is concurrent with inflations of nΨ beyond nPPV. The application of our burst detection method showed a good agreement with the ground truth in simulated data (S10 Fig). Note however that this analysis cannot exclude the contribution of other factors which may increase nΨ beyond nPPV as the correspondence between burst count and nΨ are only relative (i.e. they do not have the same units thus difference cannot be calculated). Indeed, without including the effects of all other factors on nΨ the effect of bursting cannot be readily transformed into nΨ units (S1 Appendix).

## Different schemes for coordinated activity in the presynaptic network induce independent changes in nΨ and nRV

As mentioned previously, accurate estimation of components of neural variability constrains the viable neural mechanisms that underlie the observed neural responses and can serve as a valuable tool for cross-region comparisons [4,7,45–48]. $CV^2$ of ISIs was previously reported to change as a function of correlations in pre-synaptic inputs both theoretically and using simulations [49]. However, $CV^2$ overestimates spiking irregularity in presence of rate fluctuations [30]; in contrast, as shown previously nΨ is robust to rate fluctuations (see S4 Fig).

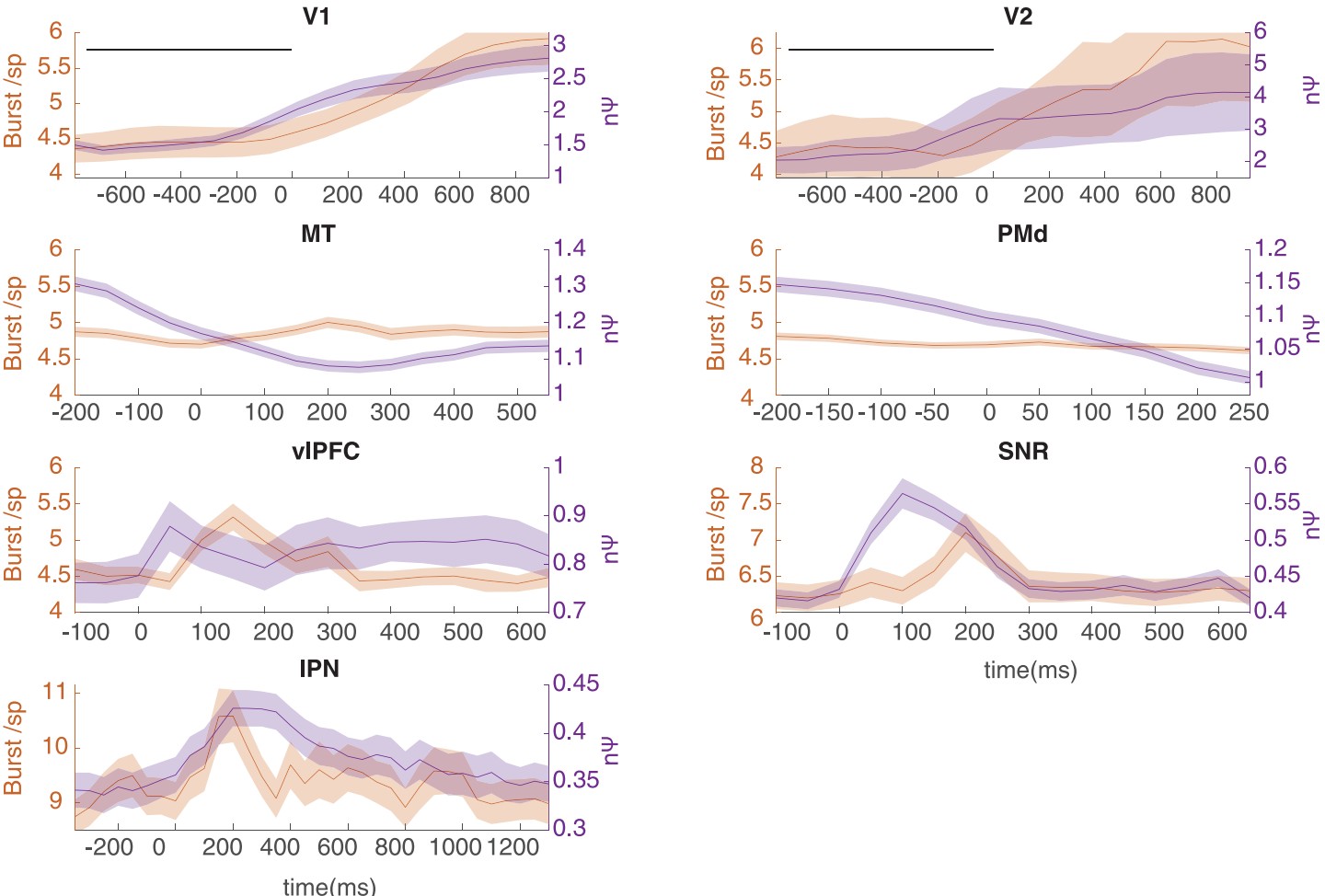

**Fig 6. Increase of nΨ is concurrent with increased bursting across some cortical and subcortical regions.** Peristimulus temporal changes in nΨ is plotted alongside temporal changes in burst count per spike across regions. Burst probability for a spike was small between 5–10% across all regions. Period of stimulus on is before zero in V1 and V2 marked by a line. Only nonresponsive neurons were used in this analysis to avoid confounding effects of firing rate changes (see S1 Text).

Specifically, we aimed to find some of the characteristics in the presynaptic network which control different sources of neural variability. This is critical since the differences in the temporal dynamics of nΨ in vlPFC compared to other cortical areas or between subcortical and cortical areas, raise questions about underlying pre- and post-synaptic factors that derive spiking in different regions. Presynaptic network characteristics of the neural models are shown to have notable impact on the post-synaptic neuron variability [7,50,51]. Rate correlation [49,52] as well as spiking synchrony [45,53–55] in presynaptic networks are among these characteristics. Moreover, there are a wealth of studies suggesting a tight balance in cortical neurons [53,56] which might also affect the variability of the post-synaptic neurons. Here, we showed how statistical properties of balanced presynaptic network can reproduce some of the patterns observed in nΨ and nRV for a simple current based leaky integrate and fire (LIF) neuron.

We first show that the degree of trial-to-trial rate correlation between balanced pools of excitatory and inhibitory neurons (E and I Pools) in the presynaptic network can result in variations in nRV with minimal changes in nΨ in the post-synaptic neuron. Here we assumed two presynaptic pools, one excitatory and the other inhibitory sending efference to one LIF neuron (Fig 7a). The response variability of the LIF neuron was examined as a function of between

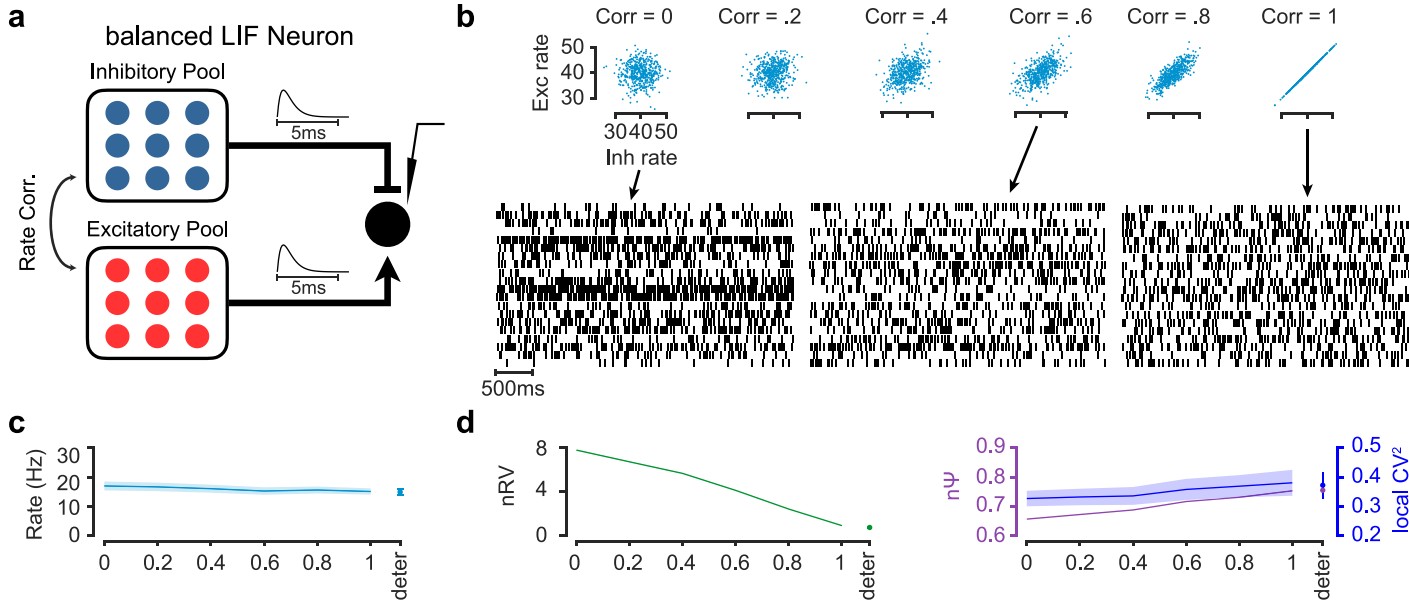

**Fig 7. Firing rate correlation between balanced presynaptic excitatory and inhibitory pools controls post-synaptic nRV.** (a) Schematic of the network model including 50 Excitatory (E pool) and 50 Inhibitory (I pool) Poisson neurons. The synaptic current is modeled using the same kernel for E and I connections. (b) E/I rate distributions of different trials (~ $N(40,4)$) with different correlation values (top) along with simulated sample rasters in three correlation conditions (correlations equal to 0, 0,6 and 1). (c) The effect of E/I rate correlation on the firing rate of post synaptic neuron. (d) The effect of E/I rate correlation on the nRV (left) and nΨ (right) of the post synaptic neuron. The effect on $CV^2_{local}$ is shown alongside nΨ. Higher rate correlation lead to lower nRV in the post-synaptic neuron but had minimal effect on nΨ or $CV^2_{local}$. Firing rate, nRV, nΨ and $CV^2_{local}$ for nonrandom presynaptic firing rate (deter) is shown alongside for comparison in c-d. Error bars and shades are sem.

pool trial-to-trial correlation in firing rate parameter (see S1 Text). Fig 7b shows the firing raster of the LIF neuron for multiple scenarios from fully uncorrelated to fully correlated firing rates (correlated $\lambda_i$ across trials $i$) between E and I pools. While the firing rate was largely unaffected by the degree of E/I pool firing rate correlation, quantitative analysis showed that indeed nRV was high for zero correlation but decreased steadily for higher correlations (Fig 7c and 7d). On the other hand, rate correlation had little effect on nΨ or nPPV (Fig 7d). The reduction of nRV as a function of rate correlation among E/I pools in the simulation is consistent with theoretical predictions (S1 Text eq.26). Further analyses, showed that increasing the balance toward excitatory pool generally results in increased firing rate and reduction of nΨ across a range of input firing rates (S11 Fig).

Next, it is shown that the degree of spiking correlation between and within E/I pools can result in reduction or enhancement of nΨ without affecting the nRV. Here, we assumed two correlated sub-pools (sub-pool #1 and #2) within each excitatory and inhibitory pools. In the first scenario, there was between pool spiking correlation for neurons in each subpool (i.e. between excitatory subpool#1 and inhibitory neurons in subpool#1, etc) (Fig 8a). In addition, a range of delays between spikes of excitatory and inhibitory neurons was considered (0, 2 and 4ms, aka tight balance [53]). The presynaptic E/I pools (50 neurons/pool) were generated using correlated Poisson processes (Macke et al., 2009). We added a Laplace distributed noise to the ISIs of the generated correlated neurons with mean zero and standard deviation of 2ms. For the correlation delay we changed the mean to 2-4ms for inhibitory neurons in order to shift inhibitory neurons after excitatory ones on average. Note that in this case the correlation exists both within each E/I subpool and between corresponding E/I subpools. Fig 8b shows simulated spiking realizations of E/I pools and the resulting spikes in the LIF neuron at 0.3

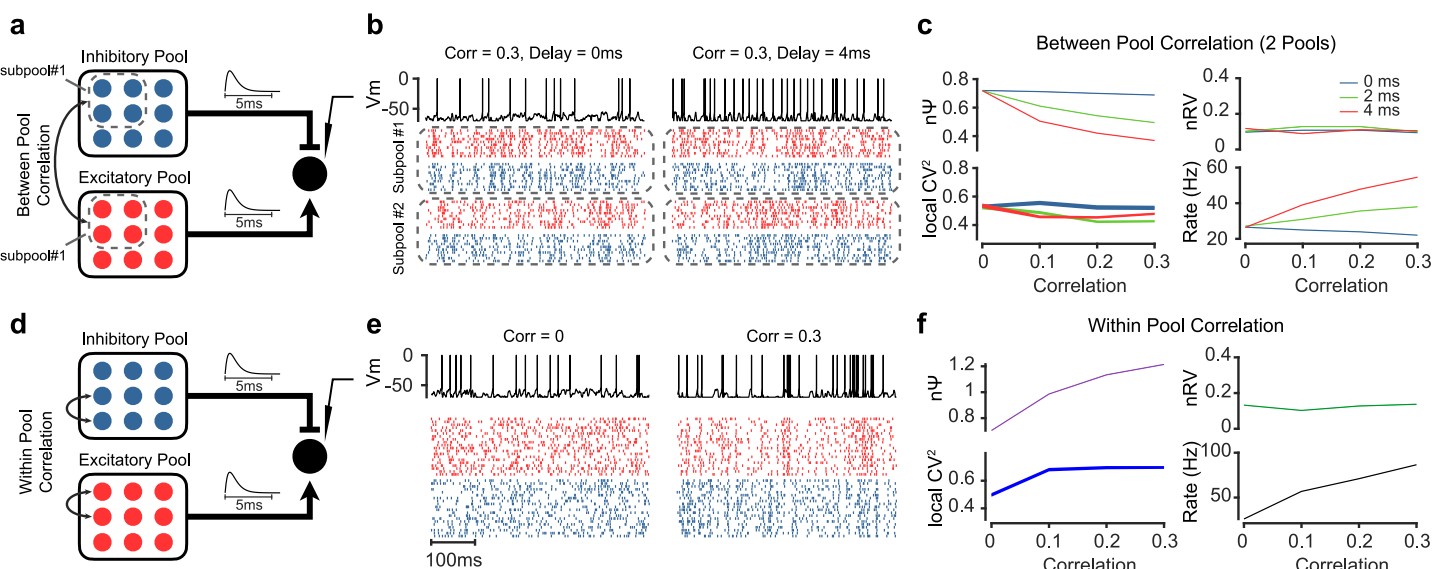

**Fig 8. Spike correlation between and within balanced presynaptic excitatory and inhibitory pools controls post-synaptic nΨ.** (a) Schematic of the network model consisting of 50 Excitatory (E pool) and 50 Inhibitory (I pool) Poisson neurons. Each pool is further divided to two sub-pools. Between pool spike correlation is imposed among sub-pools (between and within Inhibitory sub-pool#1 and excitatory subpool#2; example sub-pools are illustrated by the dotted squares). (b) Simulated spiking pattern in the E/I sub-pools 1 and 2 and the resulting spiking in the postsynaptic neuron when there is a 0.3 spiking correlation between E/I sub-pools with zero delay (left) and 4ms delay (right) between E/I neurons. (c) Stronger between pool spike correlation decreases nΨ for nonzero delays (2-4ms). nRV or $CV^2_{local}$ are less affected by between pool spike correlations. A concurrent increase in firing rate is also observed for stronger between pool spike correlation for nonzero delays (2-4ms). (d) The same network as a but with only within E and I pool spike correlation. (e) same format as b but for within pool correlations of zero (left) and 0.3 (right). (f) Stronger within pool spike correlation increases nΨ. nRV or $CV^2_{local}$ are less affected by within pool spike correlations. A concurrent increase in firing rate is also observed for stronger within pool spike correlation.

correlations [57] and 0ms (left) or 4ms (right) delays. As can be seen, the spiking irregularity was higher at 0ms compared to 4ms delay in the post-synaptic neuron (Fig 8b top). Quantitative analysis confirmed the reduction in nΨ as a function of between pool spiking correlation for nonzero delays between E/I pools (Fig 8c). This manipulation did not affect the nRV and minimally reduced $CV^2_{local}$. In this scenario the firing rate was also increased as the function of between pool spiking correlation. Similar results are found for nΨ, nRV, nPPV and rate by increasing the number of sub-pools although the effect got weaker which is expected since larger number of sub-pools result in smaller sub-pools and largely uncorrelated activity across neurons within and between pools (S12 Fig).

A different method to impose synchrony is to assume spike correlation within each E and I pools as well while excluding between pool correlation (Fig 8d). This scheme has been previously used to explain reduction of variability in response to partial inactivation of inputs from visual cortex [45]. In this case increasing the within pool spiking correlation increased spiking irregularity as shown in the example simulated rasters (Fig 8e). Consistent with previous observations high correlations lead to bouts of excitatory activity which in the absence of strong inhibitory activity, may lead to burst like activity in the post-synaptic neuron [45]. In contrast to between pool correlation, this increase in nΨ showed was concurrent with an increase in firing rate. This within pool correlation had almost no effect on nRV and resulted in a modest increase in $CV^2_{local}$.

Together these results show the plausibility of independent changes in nΨ and nRV as they were controlled by different presynaptic mechanism (rate correlation in Fig 7 vs spiking correlation in Fig 8). Notably in both cases nPPV as measured by $CV^2_{local}$ showed little changes similar to observations in the real data (Figs 3–5).

## Discussion

Neurons across the brain often show variable spiking pattern in response to the same stimulus or behavioral event [3,46,58,59]. This variability reflects both single cell level characteristics stemming from noisy sensory inputs [60], stochastic synaptic transmission [61], plasticity [62], and adaptation [63]; as well as network level properties including excitatory/inhibitory balance [64], attention and arousal levels [65], and neuromodulation effects [66]. The total variability from these sources can be formulated into two additive components: (1) variability arising from random generation of spikes given ISI parameters which we referred to as spiking irregularity (nΨ), and (2) the trial-to-trial variation in ISI parameters which in the case of renewal processes stems from rate variability (nRV). By introducing a new approach for accurate estimation of nRV and nΨ in time, we were able to reveal concurrent changes in both components across multiple subcortical and cortical regions during the stimulus presentation in comparison to the spontaneous activity. Importantly, we showed that the previously reported quench in neural variability is solely due to the reduction of nRV which was observed across all subcortical and cortical regions tested [3,4,10,17,45] (Figs 3–5). However, unlike what was previously assumed spiking irregularity was not constant but showed robust and concurrent changes with nRV. nΨ, unlike nRV, did not necessarily decrease during stimulus presentation. For instance, IpN neurons in the cerebellum and SNr neurons showed robust increases in post-stimulus nΨ (Figs 3 and 4). The increase in nΨ was also observable in the cortical neurons in the vlPFC region (Fig 5). The increase in spiking irregularity in SNr was so prominent that it canceled the reduction in rate variability resulting in the absence of the usual quench in overall variability as measured by FF.

Accurate estimation of different components of spiking variability, allows one to reveal neural strategies for encoding information beyond mean firing rate and on the decoding side opens new possibilities to decipher information hidden in neural activity about external stimuli and internal processes by an ideal observer. For instance, observing between trial variability across tasks and time periods has been discussed as another source of neural information coding, besides the averaged neural response [1,3,4,6,11,17]. Also, different patterns of neural variability in cortical and subcortical regions have been found recently [17] which may point to different encoding strategies in these regions. Subcomponents of neural variability, also, has been shown to convey information about the stimulus and neurons. For instance, nPPV has been shown to separate neurons in terms of working memory coding efficiency in prefrontal cortex [32].

nPPV, estimated by local measures of CV [23], has been previously used to explain the total variance caused by spiking irregularity [3]. However, our results showed that nPPV estimated by $CV^2_{local}$ is not able to explain the total spiking irregularity observed in neural data (nΨ). We proposed one possible mechanism that could explain this extra portion of spiking irregularity as bursting. Bursting was shown to inflate nΨ beyond predictions of nPPV (Fig 2f and S1 Appendix). In addition, between and within pool correlations in presynaptic E/I spikes were also shown to change spiking irregularity beyond inherent point process variability (Fig 8). This temporally correlated pre-synaptic activity has also been shown to be a way in which pre-synaptic activity gives rise to post-synaptic spike irregularity [45,49]. Thus nΨ estimates relative to $CV^2_{local}$ can be used to gauge the influence of sources of spike irregularity in addition to nPPV across brain regions. More research is required to both formulate the relation of these two spiking irregularity components as well as the underlying mechanisms responsible for the inflation of nΨ beyond nPPV and its possible role in neural processing and behavior.

Specifically, for IpN neurons FF showed a sustained post-stimulus increase after an initial dip which was almost concurrent with the large post stimulus surge in nΨ (Fig 3). Purkinje

cells (PC) which are presynaptic inputs to the IpN neurons are also shown to change their spiking regularity pattern in response to the stimulus onset. They also showed enhanced spiking irregularity due to inputs from molecular layer interneurons [13,39,41]. Although optogenetic manipulations in PCs have not shown sizable effects of nPPV on behavior [13], the increase in nΨ beyond nPPV (as observed Fig 3b) by mechanism such as bursting may affect behavior by increasing reliability of synaptic transmission [67]. The SNr neurons also showed increased nΨ in response to the stimulus onset independant of the excitatory or inhibitory rate changes such that post-stimulus FF remained almost constant despite the drop in nRV (Fig 4).

It is important to note that cross area comparison of spiking variability should be done with care as the meaning, the origin and the physiological function of spiking variability across different regions could be different. While spiking irregularity can be considered noise in the rate code scheme it may be the signal if a region is using temporal code. Temporal code is observed in many subcortical areas [68] including hippocampus [69], septum [70], cerebellum [71] and superior olivary nucleus [72] while at least in some cortical areas rate code may be dominant [46,73]. In addition, the role of variability in sensorimotor cortices vs associative cortices and subcortical areas may be different as we do not know or cannot always control what is considered the 'true' stimuli for such regions. In other words, spiking variability observed as nRV and/or nΨ in associative cortices and subcortical areas may well be sensitive to complex interactions between high level features of stimuli, internal states, attention and motivation that are hard to observe and control but are nevertheless different from encoding noise.

We note that the method used for estimation of variability components nRV and nΨ (Vinci or FFA) requires sufficient number of spikes per time-bin. S2 Fig shows that the need for having a high temporal resolution in estimating nRV and nΨ should be balanced by the requirement to choose a large time-bin with enough spikes ($>5$ spikes/bin) for estimation accuracy and thus may not be readily useable for regions with very low firing rates if high temporal resolution is required. Indeed, for regions with low firing rate, nRV and nΨ maybe under and over-estimated respectively for small time-bins (S2 Fig). Most of the areas examined in this study had sufficiently high firing rates but we may be prone to some misestimation for area V2 which had low firing $<5$Hz during stimulus off time (Fig 5). Note that the requirement for the sufficient number of spikes per time-bin is due to the nature of doubly stochastic point process and exists even for idealized simulations with access to a large number of trial repetitions with the same parameter (i.e. empirical estimate, S2 Fig). The problem of low spike counts can be partially addressed by using model-based approaches and additional assumption on smoothness of firing rate changes during a trial but with some loss of generality [74].

The relatively large time-bins used for estimation of nΨ may raise a concern about the changes in firing rate within a time-bin to affect the variability estimates. However, theoretical analysis as well as simulation results show that the spiking irregularity estimates are robust to such firing rate changes during the time-bin (S4 Fig and S1 Appendix). For different patterns of firing rate fluctuations (S4b–S4f Fig) in comparison to the stationary rates (S4a Fig), the $CV^2_{local}$ and nΨ remain largely unchanged. Nevertheless, in order to ensure the robustness of our main conclusions on the real neural data, we have implemented two additional strategies to control for variations in firing rate namely mean-matching and separately considering post-stimulus excitation, inhibition and null-responses for the variability analysis (S7–S9 Figs). Importantly, these additional controls resulted in virtually the same estimates of variability components across all regions.

In this work, we used LIF current based neural simulations to show differential effects of network input properties such as rate correlations and spiking correlation on nΨ and nRV. In particular higher rate correlations between excitatory and inhibitory pools were found to decrease nRV but had minimal effects on nΨ (Fig 7). On the other hand, spiking correlation within and between excitatory and inhibitory pools increased or decreased nΨ without changing nRV (Fig 8). However, conductance based models are recently gaining ground as the more viable alternative to explain synaptic effects seen in real neurons [50,75]. The choice of synaptic model (current or conductance based) affects the spiking irregularity sensitivity of the postsynaptic neuron to the presynaptic firing rate [50]. We note that our main goal in this paper was not to enumerate all possible mechanisms that control neural variability but to provide a better understanding and an accurate methodology for estimation and parsing components of variability. Obviously, while a more accurate estimation of neural response variability does not by itself identifies a unique underlying neural mechanism, it constrains possibilities that give rise to the observed temporal dynamics in each of nΨ and nRV.

In summary, our results revealed a ubiquitous quench in post-stimulus nRV (but not FF) across many cortical and subcortical regions. Importantly, we also found significant and concurrent changes in nΨ which represents spiking irregularity in all areas examined. However, unlike nRV the pattern of post-stimulus changes in nΨ was heterogenous across regions. Previous work [3] indicated a robust reduction in network noise and a constant private noise during the stimulus. Our results do not address shared and private variability components across a population. However, the fact that both nΨ and nRV components showed concurrent post stimulus changes across neurons and had plausible network level implementations (Figs 7 and 8) suggest that they contribute to network level noise. On the other hand, nPPV seemed to be resistant to change in real data and thus may represent the private noise component for each neuron [23,30]. Interestingly, previous simulations and in-vivo experiments [76] have revealed that the balanced networks of excitatory and inhibitory networks are plausible models explaining the constant nPPV phenomena as seen in our simulations. Analysis of simultaneously recorded neurons to address network level organization of nΨ and nRV across the population can further address factors underlying event related changes in neurons response variability components.

## Materials and methods

MT and PMd data (n = 7031, 4264 sessions respectively) were recorded in behaving macaques [3]. V1 and V2 datasets (n = 1272, 296 respectively) are from anesthetized macaque monkeys in 5 different sessions [77]. We used the sessions with largest number of neurons for each of these two regions and merged all different grating conditions. vlPFC (areas 8Av, 46v, and 45) and SNr data are recorded in behaving macaques [42,44] and is available at [78]. The IpN data is from deep cerebellar nucleus of mice [40] and is available at [79]. We only used the first 270 sessions which were recorded during conditioning. For all datasets, we removed neurons with very low firing rates (less than 3 spike/bin average) for the given time-bin since such low firing rates normally did not allow for accurate parsing of spike variability components (number of used sessions are indicated in Fig 5). For supplemental methods see S1 Text.

### Mean matching

Mean matching selects sub-population of neurons/conditions for each time-bin in a trial period so as to the rate distribution remains constant for all time-bins. We used the code provided by the [3] for S7–S9 Figs.

### Response polarity clustering

We used pre- and post-stimulus periods of the tasks to compute the average spike count for the two periods of each neuron. We then calculated the p-values for differences between pre- and post-stimulus rate of each neuron using paired Student T-test to group them into (1) excited (positive and $p < 0.05$), (2) null ($p > 0.05$), and (3) inhibited (negative and $p < 0.05$) groups. We then used these response types to calculate the average $n\Psi$ and $nRV$ to test if the observed changes are due to rate fluctuations (S7b–S7d and S9 Figs).

The data related to vlPFC and SNr and the code used in this work are deposited in Dryad [78].

### Dryad DOI

https://doi.org/10.5061/dryad.0cfxpnw2c

## Supporting information

**S1 Fig. Effect of rate variability vs. spiking irregularity on spiking pattern of a simulated neuron.** (a) A case where there was rate variability from 10–30 Hz before the stimulus and after 1 second from a stimulus onset at time zero. The rate variability collapsed to zero for 1 second after stimulus onset. Spiking irregularity was unchanged during this time (top) and the raster plot of simulated spike trains sorted by baseline rate from low to high (bottom). (b) A case where there was no rate variability during a trial but a reduction in spiking irregularity from 2 to 0.5 for 1 second period after stimulus onset (top) and the raster plot of some simulated trials. Spike generation was done with gamma inter-spiking interval where $\kappa$ is the shape parameter, $\theta$ is the scale parameter and firing rate $\lambda = \frac{1}{\kappa\theta}$.
(PDF)

**S2 Fig. Dependence of $n\Psi$ and $nRV$ estimates using Vinci and FFA methods on the number of trials and the average number of spikes within the time-bin.** (a) Each heat map shows estimated $n\Psi$ values for a gamma processe (10Hz) for different $nRV$ (columns) and $n\Psi$ ($\frac{1}{\kappa}$) (rows) using the Vinci method. X-axis shows average number of spikes within time-bin and y-axis shows number of trials. Color-code is adjusted such that color green means correct estimation, while red and blue mean over- and under-estimation of the true $n\Psi$ values, respectively. The heat bar over each square heatmap indicates the empirical estimates (best possible estimate given access to many repetitions of each trial). (b-d) same format as a but for estimated $n\Psi$ values using FFA, for estimated $nRV$ values using the Vinci method and for estimated $nRV$ using FFA, respectively.
(PDF)

**S3 Fig. Faster temporal fluctuations in variability favor smaller time-bins to minimize overall estimation error.** (a) Example of $nRV$ during time with a single change from high to low during the trial (black) and the estimated nRVs using the Vinci method with different sizes of time-bins. (b) The mean sum of squared error (MSE) between the estimated nRV using FFA and Vinci methods and the true nRV by varying time-bins for the nRV pattern shown in a. The optimal time-bin (250ms) resulted in the smallest MSE compared to small (150ms) or large (350ms) time-bins. The MSE for FFA was higher than Vinci but showed a similar pattern. (c) Example of a rapidly fluctuating nRV during time from high to low during the trial (black) and the estimated nRVs using Vinci method with different time-bins. (d) The optimal time-bin (190ms) resulted in the smallest MSE compared to small (150ms) or large (350ms) time-bins for the pattern shown in c. Note the smaller optimal time-bin in this example compared to a.
(PDF)

**S4 Fig. nΨ estimates are robust to changes in firing rate during the time-bin.** (a-f, top) firing rate pattern within a time-bin of 500ms for a gamma process with $\kappa = 2$. The shading shows firing rate variation across trials which should only effect nRV not nΨ, as in Fig 2c. (a-f, bottom) Empirical estimates of nΨ (red bar) and the $CV^2_{local}$ estimates (green bar) along with nΨ estimates by Vinci method.
(PDF)

**S5 Fig. VarCE can give erroneous estimations of VEC.** (a) Similar to Fig 1b example with a time varying nRV component but a constant nΨ component (constant $\kappa$) (top two rows). Empirical estimates of VEC and nΨ (same as $\phi$) along with estimates made by Vinci and VarCE methods. (b) same format as a but for a case when there is no rate variation (VEC = $T^2 \times RV = 0$) but there is time varying spiking irregularity (changing $\kappa$). VarCE method assigns fluctuations caused by nΨ to VEC while Vinci method correctly disentangles the two different sources.
(PDF)

**S6 Fig. Relative size of nΨ and nRV $\times$ $T$ in comparison to FF.** Temporal dynamics of normalized spiking irregularity nΨ and normalized rate variability nRV estimates and the fact that they sum up to almost fully explain the FF dynamics (using the FFA method). Results are shown for all subcortical and cortical regions analyzed in the main paper. The relative size and contribution of nΨ and nRV in driving the fluctuations in FF can also be examined.
(PDF)

**S7 Fig. Robustness of nΨ and nRV pattern in SNr to variations in firing rate.** (a) Mean-matching as well as clustering the SNr neurons to three groups with (b) excitatory response, (c) inhibitory response, (d) and null response yield similar patterns in nΨ and nRV estimates as shown in Fig 4. The gray curve in (a, top) plot shows the average and sem mean matched firing using a sub-selection of neurons for each time-bin. The gray curve in (b-d, top) is the PSTH evaluated with the same time-bin (200ms) as used for variability estimates nΨ and nRV.
(PDF)

**S8 Fig. Robustness of nΨ and nRV pattern across cortical regions by mean-matching.** Mean matching across cortical regions yield similar patterns in nΨ and nRV estimates as shown in Fig 5. The gray curve in the PSTH plot shows the average mean-matched firing rate.
(PDF)

**S9 Fig. Robustness of nΨ and nRV pattern across cortical regions within different response types.** Clustering cortical neurons to three groups with (a) excitatory response, (b) null response, (c) and inhibitory response yield similar patterns in nΨ and nRV estimates as shown in Fig 5.
(PDF)

**S10 Fig. Burst detection method correctly estimates bursts/sp in simulated spike trains.** (a) Sample spike rasters with two different level of bursting activity during in 0-500ms period after the stimulus onset at zero. Burst counts from a uniform distribution with mean 11 bursts per spike (top raster) or 5 bursts per spike (bottom raster) were added with 0.5 probability to normal spikes from a gamma point process with 20Hz rate and $\kappa = 2$ shape parameter. Similar to real data, method parameters were estimated based on the spontaneous activity [−500, 0] period on a pool of 20 neurons with mean firing rate of 20Hz with a baseline burst count of 3 burst/sp (b) Mean burst count /sp estimated by the method vs the actual mean burst /sp used in generating the data. The gray line shows the unity line.
(PDF)

**S11 Fig. nΨ decreases as the network goes from balanced excitation/inhibition to the one dominated by excitation.** (a) Schematic of the network model including $n_e$ Excitatory (E pool) and $n_i$ Inhibitory (I pool) Poisson neurons with rate $\lambda$. The synaptic current is modeled using the same kernel for E and I connections. (b) nΨ decreases faster as a function of presynaptic rate ($\lambda$) as the color-coded excitatory ratio $\frac{n_e}{n_e+n_i}$ approaches balanced network (i.e. 50%). (c) Postsynaptic neuron's firing rate increases faster as a function of the presynaptic firing rate when the excitatory ratio increases.
(PDF)

**S12 Fig. Effect of number of sub-pools on post-synaptic nΨ as a function of between pool correlation.** nΨ and rate sensitivity to between pool correlation decreases as we increase the number of sub-pools (for constant number of neurons). nRV and $CV^2_{local}$ remain mostly unchanged in all scenarios.
(PDF)

**S1 Appendix. Proofs for the effects of bursting and lack of an effect for firing rate fluctuations on nΨ.**
(PDF)

**S1 Text. Supplementary materials and methods.**
(DOCX)

## Acknowledgments

The authors would like to thank Hideaki Shimazaki for providing the code for simulation of spiking for general renewal processes.

## Author Contributions

**Conceptualization:** Saleh Fayaz, Ali Ghazizadeh.

**Data curation:** Mohammad Amin Fakharian, Ali Ghazizadeh.

**Formal analysis:** Saleh Fayaz, Mohammad Amin Fakharian, Ali Ghazizadeh.

**Methodology:** Saleh Fayaz, Mohammad Amin Fakharian, Ali Ghazizadeh.

**Software:** Saleh Fayaz, Mohammad Amin Fakharian, Ali Ghazizadeh.

**Supervision:** Ali Ghazizadeh.

**Visualization:** Saleh Fayaz, Mohammad Amin Fakharian, Ali Ghazizadeh.

**Writing – original draft:** Saleh Fayaz, Mohammad Amin Fakharian, Ali Ghazizadeh.

**Writing – review & editing:** Saleh Fayaz, Ali Ghazizadeh.

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
