## [Decision Letter · Decision Letter 0]

24 Nov 2021

Dear Dr. Ghazizadeh,

Thank you very much for submitting your manuscript "Stimulus presentation can enhance spiking irregularity across subcortical and cortical regions" for consideration at PLOS Computational Biology.

As with all papers reviewed by the journal, your manuscript was reviewed by members of the editorial board and by several independent reviewers. In light of the reviews (below this email), we would like to invite the resubmission of a significantly-revised version that takes into account the reviewers' comments.

We cannot make any decision about publication until we have seen the revised manuscript and your response to the reviewers' comments. Your revised manuscript is also likely to be sent to reviewers for further evaluation.

Sincerely,

Alireza Soltani

Associate Editor

PLOS Computational Biology

Lyle Graham

Deputy Editor

PLOS Computational Biology

Reviewer's Responses to Questions

**Comments to the Authors:**

Reviewer #1: 1 Overview

══════════

This paper discusses the problem of variability partitioning in neural

spike trains. It presents a method for decomposing variability in

trial spike irregularity and trial-by-trial rate variability that can

also be used in cases where the trial spike irregularity changes over

time. The method is compared with an existing approach, and together

they are applied to experimental and simulated data. The chief

conclusion from the analysis of experimental data is that external

stimuli can increase spike irregularity in some brain areas, and can

in certain cases counteract the reduction in variability that is

usually expected to follow stimulus presentation.

Overall, the conclusions from the experimental analyses presented in

this paper are interesting, and in my opinion the simulations do a

good job of providing minimal examples of mechanisms that could give

rise to certain empirical trends. However, I found the way in which

the FFA and Vinci methods are presented rather confusing or misleading

(see below). I cannot recommend the paper for publication in its

present form, but I may reconsider my recommendation if the paper was

revised to address my concerns.

2 Major concerns

════════════════

1. My main concern with this paper is that the proposed method (FFA)

seems strictly inferior to the Vinci method, except for pedagogical

purposes. If it is introduced just for the sake of explanation,

then this should be made clear from the beginning, so that the

reader doesn't go through the introduction and the initial part of

the results thinking that this is a novel contribution that solves

the problem of variability partitioning better than existing

approaches. Otherwise, if FFA is supposed to stand on its own, the

authors have to provide a clear motivation as to why a reader would

want to use it, in practice, rather than the Vinci method. At

present, such motivation seems missing from the paper.

In absence of a solid answer to this question, the reader would be

better served by a paper simply about the modified Vinci method and

its application to experimental and simulated data (and therefore

with more details on that technique, which at the moment are rather

sparse - even after reading the supplement, the reader is left with

almost no insight as to how the Vinci approach works, and how does

it look in its "modified" form), with a separate appendix or

discussion section devoted to the explanation of the FFA

perspective, corredated by a schematic like Figure 1 and some

comparisons like Figure S1 that show that this method is inferior

to Vinci. Alternatively, FFA could be removed entirely from the

paper, and published elsewhere in the context of a tutorial paper

on variability partitioning.

2. I was unable to access the data and code repository at

<https: 10.5061="" doi.org="" dryad.0cfxpnw2c="">, as the DOI does not

resolve.

3 Minor points

══════════════

1. I believe the paper would be enhanced by the addition of a

schematic with some simple raster plots (either cartoon or from a

simulation) as a figure in the introduction, showing what nΨ and

nRV differences look like visually. This would greatly enhance the

readability of the paper for readers that may not be familiar with

the general concept of variability partitioning (or that may just

need a reminder). I'm thinking of something like the raster plots

in Figure 7, which do a great job of showing the possible

components of the correlation. This is just a suggestion however.

2. When introducing the Vinci method, please use a more explicit

phrasing; "a previously introduced method" could be construed as

implying that this method is due to the same authors, when it's

not; something like "an existing method" is less ambiguous and

would work better.

4 Typos and clarity

═══════════════════

• abstract: 'aka φ' does not add anything and is quite confusing,

resulting only in added notational load for the reader. Please

consider removing.

• page 5, lines 12-13: the notation is somewhat confusing - please

annotate the expectation/variance symbols to indicate what the

expectations are taken over.

• page 14, line 23: reticula → reticulata

• page 16, figure 4 caption: repeating the caption for figure 2 would

result in better readability than writing "same format than..".

• page 16, line 13: "due to relatively large size of nΨ, dynamics of

the FF and (?) is dissimilar": likely missing symbol.

• page 17, line 22: "at the first of the trials" → "at the beginning

of the trials"

• page 20, line 14: "derive" → "drive"

• page 27, line 19: this sentence probably needs an extra comma before

"namely" for readability.

• page 28, line 9: "recently" seems odd, when used to refer to two

papers including one which is 17 years old.

• page 29, line 11: (68) seems like the wrong reference?

• page 29, lines 13-16: this sentence is also probably missing a

couple of commas, or could be improved by being broken up.</https:>

Reviewer #2: The review is uploaded as an attachment.

Reviewer #3: Fayaz et al present a high-quality paper that provides a possible solution to more accurately quantify the multiple sources of variability in single neurons’ spiking. The key issue is that variability (e.g. measured by for example Fano factors) arises due to several factors: across trial variability and ~spike irregularity. Previous results have lumped these sources of variability together, though they are different, and the information about network states that one can infer from them is distinct. I think this paper is a very good fit for PLOS COMP BIOLOGY and represents a systematic advance.

I have several conceptual suggestions which are important to consider when thinking about noise across many brain areas that should make it into the Discussion.

For a long time now, changes in variability in subcortical brain areas have been observed (often showing different changes than in "sensory cortex" cortex, (see studies of medial septum, e.g., old studies by Buzsaki showing neural rhythms for example in the septohippocampal system that occur across large groups of neurons relative to state changes). There are many reasons for the differences between variability in sensory cortex versus other areas, that I think are worth acknowledging and carefully discussing in the framing of the method.

Brain areas (cortical and subcortical) have different intrinsic timescales, may integrate information differentially relative to external stimuli (sensory events) and internal stimuli (e.g. state changes). And there broadly put: we don’t know what a “stimulus” is for many brain areas (versus V1 or A1 which have clear stimuli that can be used to systematically drive neurons and study across trial, within trial, and across neuron variability). Because we do not know what is a "stimulus" for many brain areas we do not know how to correctly interpret across trial variability - which may not be variability at all but a tight tracking of an internal state (internal stimulus) that we do not understand. Again, while we know how to drive a V1 neuron we may not really understand the high dimensional "RF" (if that is even a right concept for PFC) of a "vlPFC" neuron. So, with that in mind I think some deep conceptual limitations in across area comparisons in this and other studies of variability need to be acknowledged.

Minor

Please define vlPFC precisely. As the last author knows, vlPFC is many brain regions (eg., 46, 46, 12, etc). Please be precise in what is vlPFC.

**Have the authors made all data and (if applicable) computational code underlying the findings in their manuscript fully available?**

Reviewer #1: **No: **The DOI provided for code and data doesn't seem to resolve to anything.

Reviewer #2: **No: **The DOI provided by the authors is not valid.

Reviewer #3: Yes

PLOS authors have the option to publish the peer review history of their article (what does this mean?). If published, this will include your full peer review and any attached files.

Reviewer #1: No

Reviewer #2: No

Reviewer #3: No
---

## [Decision Letter · Decision Letter 1]

25 Mar 2022

Dear Dr. Ghazizadeh,

Thank you very much for submitting your manuscript "Stimulus presentation can enhance spiking irregularity across subcortical and cortical regions" for consideration at PLOS Computational Biology. As with all papers reviewed by the journal, your manuscript was reviewed by members of the editorial board and by several independent reviewers. The reviewers appreciated the attention to an important topic. Based on the reviews, we are likely to accept this manuscript for publication, providing that you address last concerns of Reviewer # 2 and revise the manuscript accordingly.

Sincerely,

Alireza Soltani

Associate Editor

PLOS Computational Biology

Lyle Graham

Deputy Editor

PLOS Computational Biology

[LINK]

Reviewer's Responses to Questions

**Comments to the Authors:**

Reviewer #1: I am satisfied with the authors' answer and with their effort to address my concerns.

Reviewer #2: The authors addressed some important issues, but I still have two concerns.

(1) I understand that the goal of the paper is not to enumerate the possible mechanisms that control neural variability, but to provide a more accurate method for parsing components of variability. However, the authors confirm that the purpose of parsing components is to rule out processes that cannot produce the observed dynamics in variability. Given the very large number of possible mechanisms that give rise to the same variability components, from the molecular to the cellular to the network level, can we even rule out some of the possible mechanisms? The example using a LIF neuron model provided by the authors does not answer the question of what mechanisms are ruled out. If the goal of accurately parsing variability is to infer properties of the neurons or neuronal networks, it would be important to show a concrete example in which this is possible. Alternatively, could you provide a clear explanation (including specific examples) for why is it important to refine estimates of variability components, and what can we learn from these accurate estimates?

(2) As regards point 2 in the response letter, I appreciate the clarification on how the number of spikes per time bin affects the method accuracy, but my question was how does the relative duration of the time bin compared to the duration of the trial (or the number of time bins per trial) affect accuracy? One could increase the time-bin duration to increase the number of spikes per time bin, but this would give rise to a trade-off in accuracy (as the authors also acknowledge at point 3 in the response letter), which would be useful to quantify. Specifically, can you combine errors due to (1) limited number of spikes per time bin and (2) limited number of time bins per trial into a single error estimate on the components of variability? This would provide a more rigorous method to decide the optimal time-bin duration.

Reviewer #3: thanks for addressing my concerns.

**Have the authors made all data and (if applicable) computational code underlying the findings in their manuscript fully available?**

Reviewer #1: Yes

Reviewer #2: Yes

Reviewer #3: Yes

PLOS authors have the option to publish the peer review history of their article (what does this mean?). If published, this will include your full peer review and any attached files.

Reviewer #1: No

Reviewer #2: No

Reviewer #3: No

Figure Files:

Data Requirements:

Reproducibility:

References:

---

## [Decision Letter · Decision Letter 2]

22 May 2022

Dear Dr. Ghazizadeh,

Thank you very much for submitting your manuscript "Stimulus presentation can enhance spiking irregularity across subcortical and cortical regions" for consideration at PLOS Computational Biology. As with all papers reviewed by the journal, your manuscript was reviewed by members of the editorial board and by several independent reviewers. Based on the reviews, we are likely to accept this manuscript for publication, **providing that you address the following minor issue with Data Availability and Code Sharing.**

Currently, you have shared part of the data and code in your personal website while some other part of data is available elsewhere. Based on journal policies, "All data and related metadata underlying reported findings should be deposited in appropriate public data repositories, unless already provided as part of a submitted article. Repositories may be either subject-specific repositories that accept specific types of structured data, or cross-disciplinary generalist repositories that accept multiple data types. If field-specific standards for data deposition exist, PLOS requires authors to comply with these standards. Authors should select repositories appropriate to their field of study (for example, ArrayExpress or GEO for microarray data; GenBank, EMBL, or DDBJ for gene sequences). The Data Availability Statement must list the name of the repository or repositories as well as digital object identifiers (DOIs), accession numbers or codes, or other persistent identifiers for all relevant data." **Therefore, we suggest that you deposit the new data (vlPFC and SNr data) and code in a public depository and also provide link to the rest of data  (V1, V2, MT, PMd and IpN) in the Data Availability section.**

This is a minor, but I would also suggest to not use mathematical abbreviation (nψ, φ, nRV) in the abstract to make it more inviting. The "within-trial spike regularity" and "trial-to-trial rate variability" are much more understandable.  

Sincerely,

Alireza Soltani

Associate Editor

PLOS Computational Biology

Lyle Graham

Deputy Editor

PLOS Computational Biology

[LINK]

Reviewer's Responses to Questions

**Comments to the Authors:**

Reviewer #2: The paper has improved. I think it's okay to publish it.

**Have the authors made all data and (if applicable) computational code underlying the findings in their manuscript fully available?**

Reviewer #2: Yes

PLOS authors have the option to publish the peer review history of their article (what does this mean?). If published, this will include your full peer review and any attached files.

Reviewer #2: No

Figure Files:

Data Requirements:

Reproducibility:

References:

---

## [Editor Report · Decision Letter 3]

27 May 2022

Dear Dr. Ghazizadeh,

We are pleased to inform you that your manuscript 'Stimulus presentation can enhance spiking irregularity across subcortical and cortical regions' has been provisionally accepted for publication in PLOS Computational Biology.

Best regards,

Alireza Soltani

Associate Editor

PLOS Computational Biology

Lyle Graham

Deputy Editor

PLOS Computational Biology

---

## [Editor Report · Acceptance letter]

28 Jun 2022

PCOMPBIOL-D-21-01519R3 

Stimulus presentation can enhance spiking irregularity across subcortical and cortical regions

Dear Dr Ghazizadeh,

I am pleased to inform you that your manuscript has been formally accepted for publication in PLOS Computational Biology. Your manuscript is now with our production department and you will be notified of the publication date in due course.

With kind regards,

Marianna Bach
